nanotechnology/materials science

synaptic device, non-volatile memristor, conductance modulation, spike-timing-dependent plasticity

**Authors for correspondence:**
Le Zhao
e-mail: lezhao2007@gmail.com
Qiang Li
e-mail: liqiang@qdu.edu.cn

# Synaptic memory devices from CoO/Nb:SrTiO$_3$ junction

Le Zhao[1,2], Jie Xu[3], Xiantao Shang[3], Xue Li[3], Qiang Li[3] and Shandong Li[3]

[1]School of Control Science and Engineering, Institute of Biomedical Engineering, Shandong University, Jinan, Shandong 250061, People's Republic of China
[2]Department of Human Microbiome, School of Stomatology, Shandong University, Jinan, Shandong 250012, People's Republic of China
[3]College of Physics Science, Qingdao University, Qingdao 266071, People's Republic of China

(iD) LZ, 0000-0001-7419-3835; QL, 0000-0001-8891-260X

Non-volatile memristors are promising for future hardware-based neurocomputation application because they are capable of emulating biological synaptic functions. Various material strategies have been studied to pursue better device performance, such as lower energy cost, better biological plausibility, etc. In this work, we show a novel design for non-volatile memristor based on CoO/Nb:SrTiO$_3$ heterojunction. We found the memristor intrinsically exhibited resistivity switching behaviours, which can be ascribed to the migration of oxygen vacancies and charge trapping and detrapping at the heterojunction interface. The carrier trapping/detrapping level can be finely adjusted by regulating voltage amplitudes. Gradual conductance modulation can therefore be realized by using proper voltage pulse stimulations. And the spike-timing-dependent plasticity, an important Hebbian learning rule, has been implemented in the device. Our results indicate the possibility of achieving artificial synapses with CoO/Nb:SrTiO$_3$ heterojunction. Compared with filamentary type of the synaptic device, our device has the potential to reduce energy consumption, realize large-scale neuromorphic system and work more reliably, since no structural distortion occurs.

## 1. Introduction

Synaptic electronic devices aim at emulating synaptic functions that are crucial for achieving biologically inspired neuromorphic computing [1–3]. Recently, memristors based on non-volatile resistive switching have been considered as promising candidates for synaptic device applications because they have similar transmission characteristics with biological synapses. For example, the long-term conductance changes in non-volatile memristors are similar to the permanent change of biological synaptic weight. A number of studies have been reported on employing the non-volatile memristors to replicate essential

synaptic functions such as long-term potentiation, short-term potentiation [4,5] and spike-timing-dependent plasticity (STDP) [6–10]. In these implementations, two terminals of memristors are used as the pre- and post-synaptic sites for supplying the voltage stimulations. The memristor conductance serves as a synaptic weight. And the memristor's ability to gradually change its conductance when stimulated with different input voltages is used to demonstrate the synaptic plasticity. For these implementations to be viable, various material strategies have been studied for improving the performance of the synaptic devices, such as reducing the energy consumption [11–13], pursuing better biological plausibility [14,15] and optimizing the resistive switching (RS) behaviours [16–18], etc.

Among various non-volatile memristance materials, transition metal oxides such as NiO, $Fe_2O_3$ and CoO have been widely studied due to their high resistance ratio between the high and the low resistance states, simple constituents and compatibility with complementary metal-oxide semiconductor [19–21]. In addition, the RS in these materials is generally caused by the migration of ions [22,23]. This process is suitable for mimicking the internal dynamics of the synaptic weight change, which is also related to the migration of ions between pre- and post-synaptic neurons [24,25]. However, these materials usually show limitations on the application due to state fluctuation and abrupt current leakage introduced by conductive filaments, which impose challenges in reducing energy consumption and implementing large-scale neural networks. Besides, STDP emulation, which is the basis of achieving learning and memory in artificial neural systems, has been seldom observed in these materials.

Here we have developed a memristive-based synaptic device based on $CoO/Nb:SrTiO_3(NSTO)$ heterojunction. In this structure, the memristive switching can be ascribed to the homogeneous modulation of the barrier induced by oxygen vacancies migration and charge trapping and detrapping at the heterojunction interface between the CoO layer and the $Nb:SrTiO_3$ substrate. Compared with filamentary type of the synaptic device, this heterojunction can not only reduce the energy consumption but also improve the device performance in terms of superior uniformity, reproducibility and reliability, since no structural distortion occurs. Furthermore, we demonstrate that our device intrinsically exhibits synaptic features and can be used to emulate the STDP learning rule.

# 2. Material and methods

Heterojunctions were prepared by DC magnetron sputtering with a base pressure of $5 \times 10^{-8}$ Torr at room temperature. A schematic of heterojunction $CoO/Nb:SrTiO_3$ with a junction area of $0.5 \times 0.5$ mm is shown in figure 1a. The polycrystalline CoO layer with a thickness of 50 nm was deposited on $Nb:SrTiO_3$ substrate through shadow masks by DC reactive sputtering of Co target under an argon–oxygen mixture atmosphere of $6 \times 10^{-3}$ Torr with 10% oxygen ratio. Then, 100 nm thick Au top electrodes were deposited on CoO films by thermal evaporation. Electrical characteristics of $CoO/Nb:SrTiO_3$ heterojunctions were measured using a Keithley 2400. Voltage pulses that were used to stimulate the device were applied by a data acquisition card (NI PCI6259) controlled by a computer. During the measurement, positive bias refers to the current flowing from the top CoO electrode to the bottom $Nb:SrTiO_3$ substrate.

# 3. Results and discussion

Figure 1a shows the schematic of heterojunction $CoO/Nb:SrTiO_3$. Figure 1b presents the current–voltage (IV) characteristics of CoO/NSTO in semilogarithmic scale. It is measured by applying a sweeping voltage of $0\,V \rightarrow +1.5\,V \rightarrow 0\,V \rightarrow -2\,V \rightarrow 0\,V$. The arrows in figure 1b indicate the sweeping direction. The device resistance can be set to high-resistance state (HRS) to low-resistance state (LRS) under a positive voltage and reset to HRS under a negative voltage. Besides, the device shows a typical non-volatile bipolar memristive switching performance without a forming process. The memristive characteristic is suitable for mimicking biological synaptic weight modulation for two reasons. First, the resistance variation induced by voltage stimuli is similar to the synaptic weight modification responding to the firing of pre- and post-synaptic neurons. Second, bipolar switching avoids abrupt resistance transition, so that it supports analogue-like transition between the resistance states. This is the base of achieving continuous change in synaptic weight. To investigate the origin of the switching behaviour, we also carried out nonlinear fittings to the transport characteristics. Figure 1c shows a $\ln(I/V)$ versus $V^{1/2}$ plot, indicating the Poole–Frenkel (P–F) emission mechanism at HRS. However, figure 1d shows an approximate linear relationship between $\ln(I)$ and $\ln(V)$, showing space charge limited (SCL) conduction at LRS. This behaviour is consistent with previous reports on the interfacial effects, where the migration of oxygen vacancies and charge trapping/detrapping in the vicinity of

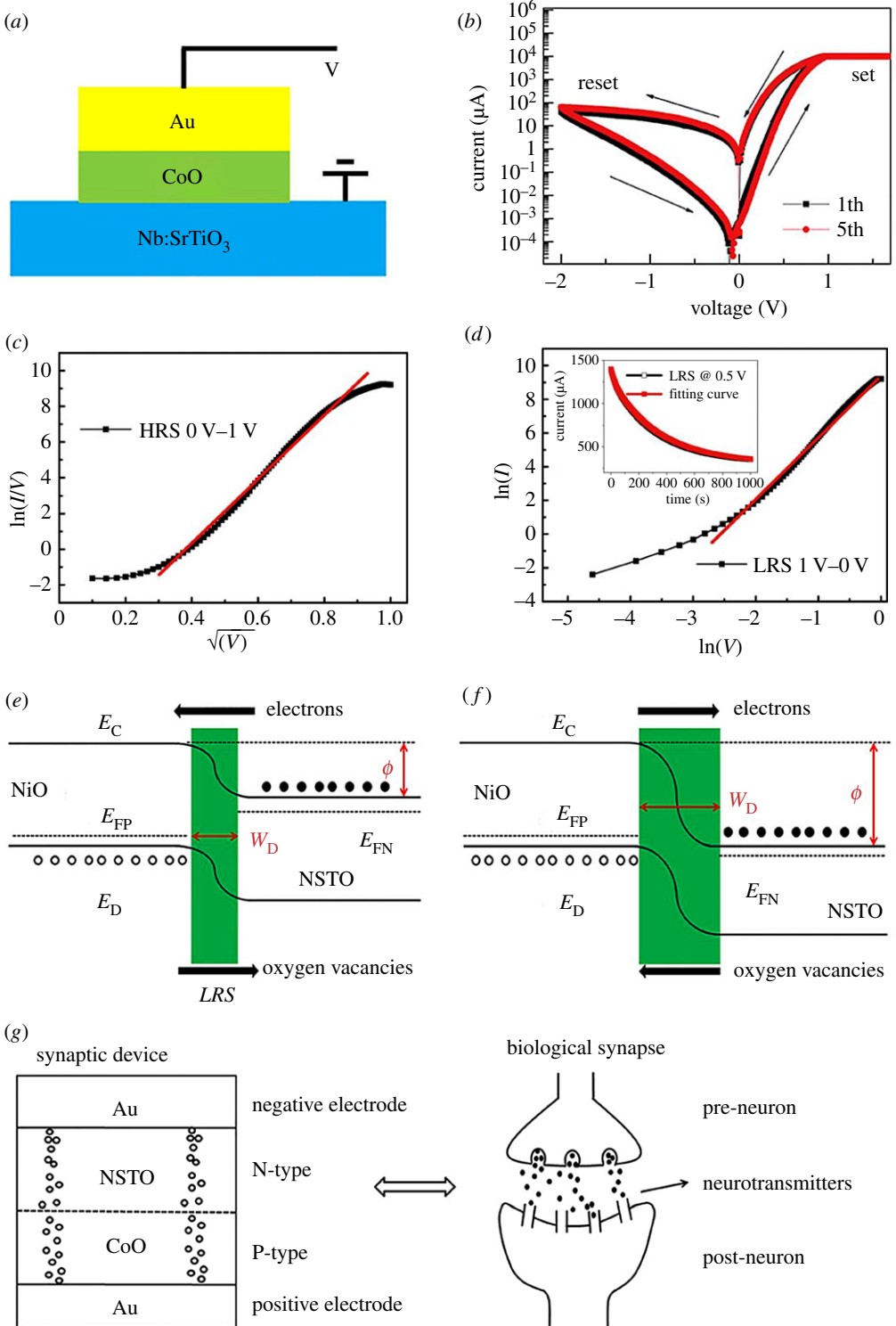

**Figure 1.** (*a*) The schematic of heterojunction CoO/Nb:SrTiO$_3$. (*b*) *I–V* characteristic showing non-volatile bipolar memristive switches. Arrows indicate the voltage-sweep direction. Five cycles of operation are shown. (*c*) ln(*I/V*) versus (*V*)$^{1/2}$ in the negative bias region at HRS. (*d*) ln(*I*) versus ln(*V*) in the negative bias region at LRS. Inset: the time dependence of the current at the LRS and the fitting curve. (*e,f*) The schematic diagram of the mechanism of RS behaviours in CoO/Nb:SrTiO$_3$ heterostructure. (*e*) At forward bias voltage (LRS) and (*f*) at reverse bias voltage (HRS). (*g*) Analogy between the CoO/Nb:SrTiO$_3$-based device and the biological synapse.

the interface drives RS in various heterojunctions [26–28]. Moreover, the time dependence of the current at the LRS was fitted with a bi-exponential relaxation equation of $I = I_0 + Ae^{-t/\tau_1} + Be^{-t/\tau_2}$, where $I_0$ is the steady-state current, $t$ is the time, A and B are constant, and $\tau_1$ and $\tau_2$ are two relaxation time constants.

The well-fitted curve with $\tau_1 = 27.7$ s and $\tau_2 = 337.6$ s further implies oxygen vacancy migration. Thus, the memristive switching here can be ascribed to the modulation of the barrier via the migration of oxygen vacancies and charge trapping and detrapping at the heterojunction interface between the CoO layer and the Nb:SrTiO$_3$ substrate [27,28]. This is illustrated in figure 1e. When a positive voltage is applied to the CoO electrode, the oxygen vacancies would migrate toward the NSTO electrode and assemble at the CoO/NSTO interface. Meanwhile, the electrons in the NSTO would move toward the CoO electrode and become trapped by the oxygen vacancies. The depletion layer width becomes narrower, and the interface barrier height becomes lower, which makes it easier for the electrons to cross the barrier into the CoO film. When the reverse voltage is applied, as shown in figure 1f, electrons trapped by the oxygen vacancies would be released and migrate toward the NSTO electrode, while the oxygen vacancies would migrate in the CoO direction. The depletion layer becomes wider and the barrier height becomes higher, and then the device resistance state would switch from LRS to HRS.

Figure 1g illustrates an analogy between the electronic synapse based on CoO/Nb:SrTiO$_3$ memristor and the biological synapse. Note that in this paper, we specifically focus on emulating excitatory synapses because phenomena, such as long-term potentiation and long-term depression, usually occur at these synapses [29]. As shown in figure 1g, a biological synapse is composed of a pre-synaptic neuron, a post-synaptic neuron and a synapse. In electronic synapse based on CoO/Nb:SrTiO$_3$ memristor, the negative and positive electrodes work as the pre- and post-synaptic neurons, respectively. The synaptic weight, which reflects the connection strength between two neighbouring neurons, is represented by the device resistance or conductance. The resistance or conductance of the electronic device is changed because of the oxygen vacancies migration when receiving the input action potentials. This is similar to the biological synaptic weight, which is changed because the pre-neuron releases ions when receiving the input spikes. And the device resistance is inversely proportional to the synaptic weight while the device conductance is proportional to the synaptic weight.

In practical operation, applications of synaptic devices are under pulse signal inputs rather than the voltage ramp in a DC sweep. Thus, we investigated the property of resistivity switching of our device in the context of programmed voltage pulses. We set the pulse width at 2 ms because the biological spikes usually last for 1–2 ms [30]. The device resistance was monitored by a read voltage of 100 mV for 100 ms after each pulse stimulus.

Figure 2a shows the resistance shifts when a single positive square pulse with different voltage amplitudes is applied to the device. It can be seen that the SET (high to low resistance change) process has a threshold bias level around $V_{th} \sim 0.8$ V. This matches with the BCM rule for biological synapse [31]. The BCM rule indicates that there is a threshold value of external stimulation for long-term potentiation induction. In the biological synapse, the 'threshold' is attributed to voltage-gated channels, which controls the release of ions. In our device, the threshold results from the energy accumulation for the migration of oxygen vacancies and charge trapping. As indicated in figure 2a, pulses with bigger amplitudes lead to a bigger shift in resistance. This also happens to biological synaptic weight. The amount of variation in biological synaptic weight is proportional to the amplitudes of stimulation spikes.

Because the variation amount accumulates if the stimulus is sustained in synapses [32], we further investigate the memristive properties of CoO/NSTO devices by applying sequences of pulses. Each sequence of pulses has a specific amplitude and a fixed width of 2 ms. The time intervals between pulses were 1 s (900 ms waiting time plus 100 ms read time). During this interval, the resistance state was monitored by a read voltage of 100 mV. The time interval between pulses would not affect the device resistance because of the non-volatile memory of the device, which refers to a stable resistance state after being stimulated. The initial $R_{HRS}$ state was selected as approximately 130 kΩ. Figure 2b displays the measured resistance as a function of pulse number for SET process as well as for different pulse amplitudes. The device exhibits gradual changes of resistance following the pulse train. The response of our device to the pulse train is functionally similar to long-term potentiation properties in biological synapses [33]. This property defines synaptic plasticity, which is the ability of synapses to change their strength. Synaptic plasticity is believed to be the major cellular mechanism underlying learning and memory. Moreover, both speed and magnitude of the resistive switching increase with the pulse amplitude, well mimicking the faster response of synapse to enhanced neurostimulation. In addition, when sustained spikes are applied to the device, the changing rate of the resistance decreases, and the resistance finally reaches an upper limit, which is higher with larger pulse amplitude. This is consistent with biological phenomena. Synaptic modification is most pronounced early in the learning process, and synaptic weight is reinforced as the learning process continues. However, the synaptic weight becomes saturated when it reaches a limited value even under continuous stimulation [34]. This

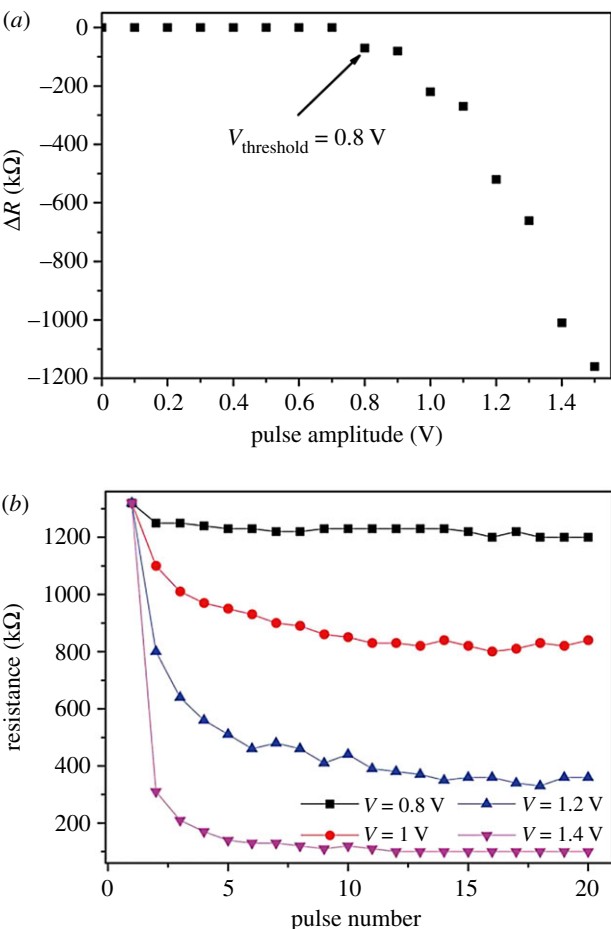

**Figure 2.** (a) Resistance shift in CoO/NSTO device when a single positive pulse with varying amplitudes and fixed duration of 2 ms is applied to the device. (b) Dependence of the device resistance on the pulse amplitude and pulse number.

is for avoiding uncontrolled growth of synaptic strength and preventing excessive neural firing [35]. Physically, the nonlinear relationship between the resistivity and the switching pulse number means that the response of our device to the particular electrical pulse depends on the current resistance state. The observed resistance evolution is related to the establishment of new potential balance caused by the migration of oxygen vacancies and charge trapping.

Next, we implement a more accurate modification of synaptic efficacy. Biological synaptic weight can be accurately increased or decreased when they are alternately stimulated by potentiating or depressing spikes, respectively. To emulate this synaptic weight modification, we used positive and negative square pulses with varying amplitudes to represent potentiating and depressing spikes, respectively, and test the continuous resistance programmability of our device. All pulse widths were fixed at 2 ms. Each pulse was followed by measurement of the device resistance. Note that the time interval between pulses would not affect the device resistance because of the non-volatile memory of the device. First, the device was set to HRS. Figure 3 shows a gradual decrease of our device resistance by using positive voltage pulses with increasing voltage amplitude from 0.8 to 2 V with a 0.1 V step and subsequent gradual increase in the resistance by using negative pulses with increasing voltage amplitude from $-0.8$ to $-2$ V with a $-0.1$ V step. The resistance of our device exhibited a gradual and repeatable change ranging from 50 to 1490 kΩ, and the mean resistance change per pulse was approximately 120 kΩ, which indicates a relative accuracy of 8%. The result indicates that analogous resistance programmability can be achieved in our device by changing polarities of pulse stimulation and fine-tuning the voltage amplitude. Here, the stable, accurate and repeatable resistance tuning results from the modulation of barrier in the material, which can avoid the disorganization of the material. The result in figure 3 encouraged us to implement more complex synaptic functionalities in the CoO/NSTO device.

STDP is a sophisticated synaptic modulation that the precise relative timing of pre- and post-synaptic spikes determines the sign and magnitude of the long-term synaptic weight change. It is considered to be

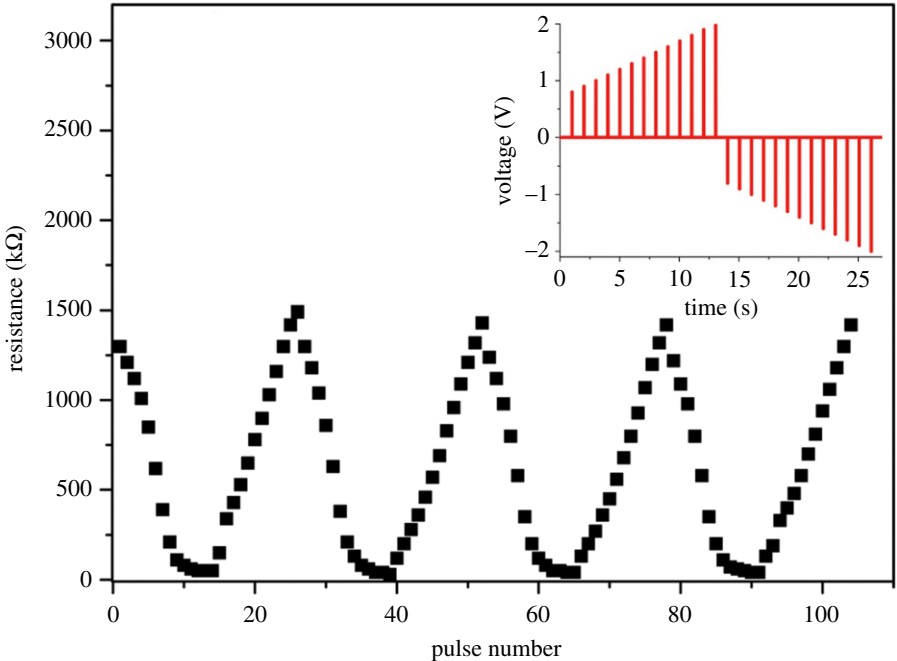

**Figure 3.** Resistance switching cycles driven by consecutive positive or negative pulses, representing synaptic weight modulation due to the potentiating or depressing pulses, respectively. Upper inset: the pulse schemes. The pulse amplitudes vary with identical 2 ms widths and 1 s pulse intervals.

a possible first law of synaptic plasticity. STDP has been found to vary in different cases [36]. In this paper, we emulate the asymmetric Hebbian learning rule [37]. According to the rule, the synapse potentiates if a pre-synaptic spike precedes a post-synaptic spike ($\Delta t > 0$) and the synapse depresses if a post-synaptic spike precedes a pre-synaptic spike ($\Delta t < 0$). Generally, a smaller spike-timing difference leads to a larger synaptic modification. Here, we use the conductance change of the device to represent the synaptic weight modulation. Although the conductance is always positive, the change of it can either be positive or negative.

In order to demonstrate the STDP function, we employ a spike pair protocol (shown in the inset in figure 4), which is also adopted in other electronic synapses [38]. The detailed analysis of the different shapes of 'spikes' and the resulting STDP function was discussed in the paper of Linares-Barranco & Serrano-Gottaredona [39]. Although the shape of the spike is not decisive, it is assumed that each spike in the pair consists of a narrow short positive pulse of large amplitude and a long relaxing slowly decreasing negative tail. In order to exclude conductance modification by a single spike, the positive pulse and tail voltage amplitudes were selected to be both lower than the threshold voltage $V_{th}$ for SET and RESET processes. And we arbitrarily set the duration of the short positive pulse and the duration of the tail voltage to be 50 and 380 ms, respectively. In this way, the spike has the shape very close to those in biological systems. Meanwhile, this setting leads to stimulation that is great enough to induce the conductance variation. In sum, the particular parameters of spikes in our experiments are as follows: $V = 0.7$ V, $t = 50$ ms for the positive short pulse, and $V = -0.3$ V, $t = 380$ ms for the following negative tail, as shown in the inset in figure 4. In practical operation, both of the two spikes were injected into the positive electrode of the device with a time difference of $\Delta t$. The two spikes were equal in magnitude but opposite in voltage polarity. The negative electrode of the device was grounded. When the two spikes with opposite polarities arrived at the device, voltages added up. The amplitude of the total voltage at the device could exceed the threshold voltage $V_{th}$, leading to a conductance variation. As $\Delta t$ increases, both the effective voltage and the effective time decrease, eventually leading to the decrease in synaptic weight change, and vice versa.

Figure 4 shows the resulting timing-dependent conductance modulation. The relative change of the conductance as a function of the spike delay time $\Delta t$ is consistent with STDP function. For high $\Delta t$, the magnitude of injected voltage does not exceed $V_{th}$ at any time, thus conductance remains unchanged. For low $\Delta t$, however, the magnitude of injected voltage in excess of $V_{th}$ is applied and the conductance is thus increased or decreased depending on the sign of $\Delta t$. The conductance change reached 14.8% for positive $\Delta t$ and −7.0% for negative $\Delta t$. Figure 4 demonstrates a successful STDP emulation. It results from the

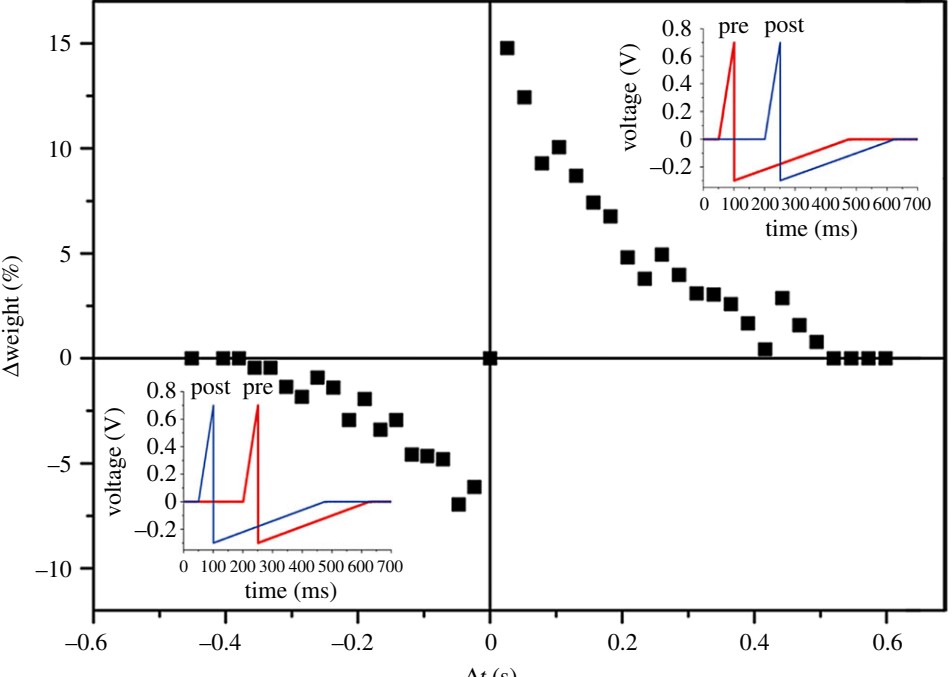

**Figure 4.** Plot of the device conductance (synaptic weight) change with variation in $\Delta t$ showing the STDP response of the electronic synapse. Insets: schematic of the pulse-pair applied to our device for STDP demonstration. Although the amplitude of each pulse is under the threshold, the sum of them can exceed the threshold of SET and RESET process and lead to change in device conductance (synaptic weight). When the pre-spike precedes the post-spike ($\Delta t > 0$), the device is set (potentiated); when the post-spike precedes the pre-spike ($\Delta t < 0$), the device is reset (depressed). The timing ($\Delta t$) between the two spikes determines the voltage flux ($V \times \Delta t$) across the device.

non-filamentary conduction of our device, which can avoid the abrupt characteristics and realize gradual conductance modulation. In addition, it is worth noting that the synaptic weight change shown in figure 4 is asymmetric. This is in accordance with the asymmetry of the $I-V$ curves with respect to bias polarity, as shown in figure 1$b$. This phenomenon can be explained by the asymmetry band barrier at the heterojunction interface between the CoO layer and the Nb:SrTiO$_3$ substrate.

## 4. Conclusion

Non-filamentary RS behaviours were found in CoO/NSTO films. The intrinsic transport property of the film is analogous to biological synaptic features. And gradual conductance modulation was realized by regulating the amplitude and polarity of stimulation spikes. Based on the conductance modulation, STDP learning rule was implemented.

Data accessibility. Data available from the Dryad Digital Repository: https://doi.org/10.5061/dryad.7c41q4c [40].

Authors' contributions. L.Z. and Q.L. conceived the experiment and wrote the manuscript. X.S. and X.L. performed the experiment in close collaboration with the rest of the authors. All the authors have contributed to data analysis and review of the manuscript.

Competing interests. We declare we have no competing interests.

Funding. This work was supported by the International Postdoctoral Exchange Fellowship Program of Shandong University, National Natural Science Foundation of China (11504192, 11674187 and 11604172) and an Innovation project of Qingdao (16-5-1-2-jch).

Acknowledgements. The authors thank Shuyun Yu and the other members in Spintronics group, Shandong University for helpful discussions.

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
