## [Reviewer comments · Royal Society Open Science]

Review History

RSOS-181098.R0 (Original submission)

Review form: Reviewer 1

Is the manuscript scientifically sound in its present form?

Yes

Are the interpretations and conclusions justified by the results?

No

Is the language acceptable?

Yes

Is it clear how to access all supporting data?

Not Applicable

Do you have any ethical concerns with this paper?

No

Have you any concerns about statistical analyses in this paper?

No

Recommendation?

Accept with minor revision (please list in comments)

Comments to the Author(s)

This experimental paper describes a novel type of memristive pn junction which emulates the conductance dependence of STDP synapses. The approach is original as it proposes a novel trapping mechanism which differs from that of Magneli alloys. The methodology appears sounds overall but has some gaps in logic which the authors ought to address before the paper is published:

1. While the mechanism of migration of Oxygen vacancies is plausible, its effect on the current through the pn junction ought to be clarified. Either vacancies change the density of minority carriers by increasing the barrier height in which case their effect should be discussed in relation to the Schokley equation for the pn junction OR the current increases through the formation of deep O₂ levels in the gap of the pn junction increasing the leakage current. Which of these two effects is involved? Why is one material p-doped and the other n-doped?

2. In multiple places in the paper there is confusion between "current", "conductance", "resistance" and "synaptic weight". For example in Fig.1 when $V > 0$ the current increases with V but the conductance decreases (due to the sub-linear current responses). Another example in Fig.4, the relation between "weight" and "conductance" should be clarified as the conductance is always positive but the weight is negative. Please define what you mean by weight.

Minor comments:

- A horizontal timescale in the inset of Fig.3 would be useful.
- The description of voltage pulses used to demonstrate STDP action in Fig.4 is not described sufficiently clearly in the text or the caption to allow the experiment for be replicated. As these are not square pulses, explain what the $t=50\text{ms}$ and $t=380\text{ms}$ mean.

Review form: Reviewer 2**Is the manuscript scientifically sound in its present form?**

Yes

Are the interpretations and conclusions justified by the results?

No

Is the language acceptable?

Yes

Is it clear how to access all supporting data?

No

Do you have any ethical concerns with this paper?

No

Have you any concerns about statistical analyses in this paper?

I do not feel qualified to assess the statistics

Recommendation?

Major revision is needed (please make suggestions in comments)

Comments to the Author(s)

Le Zhao et al investigated resistance switching properties of CoO/Nb:SrTiO₃ junctions and found that the pulse-voltage-induced resistance changes of the CoO/Nb:SrTiO₃ junctions could emulate properties of synapse, proposing the CoO/Nb:SrTiO₃ junctions as artificial synapse. However the reviewer was not convinced with conclusions and interpretations of their experimental data as described below. The reviewer would not recommend the manuscript for publication until the issues raised by the author is clarified.

(1) It is said that the migration of oxygen vacancies and charge trapping and detrapping at the heterojunction interface, which modify the depletion layer width and the interface barrier height, is responsible for the resisting switching behavior in the CoO/Nb:SrTiO₃ junctions. The reviewer was not sure from which experimental data the author concluded this proposed mechanism.

(2) The schematic in Fig 1e indicates that oxygen vacancies could move from CoO layer' surface on the positive electrode side to the Nb-STO layer's surface on the negative electrode. Is there any experimental evidences on the oxygen ion movements in the CoO/Nb-STO junctions? The reviewer was not sure how the authors verify the schematic. More explanations will be necessary.

(3) The authors need to add more explanations on details of their measurements. Especially measurements of the junction resistance changes induced by applying pulse voltages. How did the authors measure/probe the resistance changes after applying pulse voltages?

(4) For measurements in Figures 2 and 3, how long intervals between pulses were used? If the interval would be changed, how would the junction resistance behavior be influenced? The authors are asked to add more descriptions. If it is the case that the intervals affect the junction resistance behavior, is it still safely say that the Co/Nb-STO junction can emulate synapse properties?

Decision letter (RSOS-181098.R0)

18-Oct-2018

Dear Dr Zhao,

The editors assigned to your paper ("Synaptic memory devices from CoO/Nb:SrTiO₃ junction") have now received comments from reviewers. We would like you to revise your paper in accordance with the referee and Associate Editor suggestions which can be found below (not including confidential reports to the Editor). Please note this decision does not guarantee eventual acceptance.

Please submit a copy of your revised paper before 10-Nov-2018. Please note that the revision deadline will expire at 00.00am on this date. If we do not hear from you within this time then it will be assumed that the paper has been withdrawn. In exceptional circumstances, extensions may be possible if agreed with the Editorial Office in advance. We do not allow multiple rounds of revision so we urge you to make every effort to fully address all of the comments at this stage. If deemed necessary by the Editors, your manuscript will be sent back to one or more of the

original reviewers for assessment. If the original reviewers are not available, we may invite new reviewers.

- Data accessibility

If you wish to submit your supporting data or code to Dryad (<http://datadryad.org/>), or modify your current submission to dryad, please use the following link:
<http://datadryad.org/submit?journalID=RSOS&manu=RSOS-181098>

- Competing interests

- Authors' contributions

- Acknowledgements

- Funding statement

Please note that Royal Society Open Science charge article processing charges for all new submissions that are accepted for publication. Charges will also apply to papers transferred to Royal Society Open Science from other Royal Society Publishing journals, as well as papers submitted as part of our collaboration with the Royal Society of Chemistry (<http://rsos.royalsocietypublishing.org/chemistry>). If your manuscript is newly submitted and subsequently accepted for publication, you will be asked to pay the article processing charge, unless you request a waiver and this is approved by Royal Society Publishing. You can find out more about the charges at <http://rsos.royalsocietypublishing.org/page/charges>. Should you have any queries, please contact openscience@royalsociety.org.

on behalf of Dr Chong Li (Associate Editor) and Prof. Miles Padgett (Subject Editor)
openscience@royalsociety.org

Associate Editor's comments (Dr Chong Li):

Dear Authors,

Thank you for submitting your work to Royal Society Open Science for consideration of publishing. Based on the feedback from our fellow reviewers, we cannot accept your paper unless all major concerns raised by our reviewers have been satisfactorily addressed.

Please consider all reviewers' comments carefully and address all points raised by the reviewers in the revised manuscript, especially the second reviewer's concerns on some claims you made and methodologies you implemented.

Yours sincerely

Associate Editor
Dr. Chong Li

Reviewers' Comments to Author:

Reviewer: 1

Comments to the Author(s)

This experimental paper describes a novel type of memristive pn junction which emulates the conductance dependence of STDP synapses. The approach is original as it proposes a novel trapping mechanism which differs from that of Magneli alloys. The methodology appears sounds overall but has some gaps in logic which the authors ought to address before the paper is published:

1. While the mechanism of migration of Oxygen vacancies is plausible, its effect on the current through the pn junction ought to be clarified. Either vacancies change the density of minority carriers by increasing the barrier height in which case their effect should be discussed in relation to the Schokley equation for the pn junction OR the current increases through the formation of deep O₂ levels in the gap of the pn junction increasing the leakage current. Which of these two effects is involved? Why is one material p-doped and the other n-doped?
2. In multiple places in the paper there is confusion between "current", "conductance", "resistance" and "synaptic weight". For example in Fig.1 when $V > 0$ the current increases with V but the conductance decreases (due to the sub-linear current responses). Another example in Fig.4, the relation between "weight" and "conductance" should be clarified as the conductance is always positive but the weight is negative. Please define what you mean by weight.

Minor comments:

- A horizontal timescale in the inset of Fig.3 would be useful.
- The description of voltage pulses used to demonstrate STDP action in Fig.4 is not described sufficiently clearly in the text or the caption to allow the experiment for be replicated. As these are not square pulses, explain what the $t=50\text{ms}$ and $t=380\text{ms}$ mean.

Reviewer: 2

Comments to the Author(s)

Le Zhao et al investigated resistance switching properties of CoO/Nb:SrTiO₃ junctions and found that the pulse-voltage-induced resistance changes of the CoO/Nb:SrTiO₃ junctions could emulate properties of synapse, proposing the CoO/Nb:SrTiO₃ junctions as artificial synapse. However the reviewer was not convinced with conclusions and interpretations of their experimental data as described below. The reviewer would not recommend the manuscript for publication until the issues raised by the author is clarified.

- (1) It is said that the migration of oxygen vacancies and charge trapping and detrapping at the heterojunction interface, which modify the depletion layer width and the interface barrier height, is responsible for the resisting switching behavior in the CoO/Nb:SrTiO₃ junctions. The reviewer was not sure from which experimental data the author concluded this proposed mechanism.
- (2) The schematic in Fig 1e indicates that oxygen vacancies could move from CoO layer' surface on the positive electrode side to the Nb-STO layer's surface on the negative electrode. Is there any experimental evidences on the oxygen ion movements in the CoO/Nb-STO junctions? The reviewer was not sure how the authors verify the schematic. More explanations will be necessary.
- (3) The authors need to add more explanations on details of their measurements. Especially measurements of the junction resistance changes induced by applying pulse voltages. How did the authors measure/probe the resistance changes after applying pulse voltages?

(4) For measurements in Figures 2 and 3, how long intervals between pulses were used? If the interval would be changed, how would the junction resistance behavior be influenced? The authors are asked to add more descriptions. If it is the case that the intervals affect the junction resistance behavior, is it still safely say that the Co/Nb-STO junction can emulate synapse properties?

Author's Response to Decision Letter for (RSOS-181098.R0)

See Appendix A.

RSOS-181098.R1 (Revision)

Review form: Reviewer 1

Is the manuscript scientifically sound in its present form?

Yes

Are the interpretations and conclusions justified by the results?

Yes

Is the language acceptable?

Yes

Is it clear how to access all supporting data?

Not Applicable

Do you have any ethical concerns with this paper?

No

Have you any concerns about statistical analyses in this paper?

No

Recommendation?

Accept as is

Comments to the Author(s)

The authors have carefully considered and conscientiously answered reviewer comments. The paper may be published as is.

Review form: Reviewer 2

Is the manuscript scientifically sound in its present form?

No

Are the interpretations and conclusions justified by the results?

No

Is the language acceptable?

Yes

Is it clear how to access all supporting data?

Not Applicable

Do you have any ethical concerns with this paper?

No

Have you any concerns about statistical analyses in this paper?

I do not feel qualified to assess the statistics

Recommendation?

Major revision is needed (please make suggestions in comments)

Comments to the Author(s)

The reviewer thanks authors for their efforts on revising the manuscript and responding comments/suggestions. However, the reviewer is still not sure if migrations of oxygen vacancies are the main origin for the resistance switching behavior of the CoO/Nb-STO junctions.

Although the authors are claiming that this conclusion is supported by the non-linear analysis on the IV characteristics of the junctions, the authors' analysis is indirect and the oxygen vacancy migration scenario is just one of possibilities. This is a critical point because some discussion in this manuscript are based on this claim. The reviewer thinks that the author should provide direct evidences on this point.

Decision letter (RSOS-181098.R1)

12-Feb-2019

Dear Dr zhao:

Manuscript ID RSOS-181098.R1 entitled "Synaptic memory devices from CoO/Nb:SrTiO₃ junction" which you submitted to Royal Society Open Science, has been reviewed. The comments of the reviewer(s) are included at the bottom of this letter.

Please submit a copy of your revised paper before 07-Mar-2019. Please note that the revision deadline will expire at 00.00am on this date. If we do not hear from you within this time then it will be assumed that the paper has been withdrawn. In exceptional circumstances, extensions may be possible if agreed with the Editorial Office in advance.

We do not generally allow multiple rounds of revision, but the Editors have made an exception in this case.

No further rounds of revision will be possible, so we urge you to make every effort to fully address all of the comments at this stage. If deemed necessary by the Editors, your manuscript will be sent back to one or more of the original reviewers for assessment. If the original reviewers are not available we may invite new reviewers.

- Ethics statement

- Data accessibility

- Competing interests

- Authors' contributions

- Acknowledgements

- Funding statement

Kind regards,

Andrew Dunn

on behalf of Dr Chong Li (Associate Editor) and Miles Padgett (Subject Editor)

Associate Editor Comments to Author (Dr Chong Li):

Dear Authors,

Thank you for revising the paper after considering our fellow reviewers' comments. However one of our reviewers is still concerned about the real mechanism of the resistance switching phenomenon. Although the authors tried to use curve-fitting to explain it with relevant reference support in the reply, it is still not convincing. It would be better if this can be experimentally demonstrated as shown in Ref 26.

Yours sincerely

Dr. Chong Li

Associate Editor

Reviewer comments to Author:

Reviewer: 1

Comments to the Author(s)

The authors have carefully considered and conscientiously answered reviewer comments. The paper may be published as is.

Reviewer: 2

Comments to the Author(s)

The reviewer thanks authors for their efforts on revising the manuscript and responding comments/suggestions. However, the reviewer is still not sure if migrations of oxygen vacancies are the main origin for the resistance switching behavior of the CoO/Nb-STO junctions.

Although the authors are claiming that this conclusion is supported by the non-linear analysis on the IV characteristics of the junctions, the authors' analysis is indirect and the oxygen vacancy migration scenario is just one of possibilities. This is a critical point because some discussion in this manuscript are based on this claim. The reviewer thinks that the author should provide direct evidences on this point.

Author's Response to Decision Letter for (RSOS-181098.R1)

See Appendix B.

Decision letter (RSOS-181098.R2)

21-Mar-2019

Dear Dr Zhao,

I am pleased to inform you that your manuscript entitled "Synaptic memory devices from CoO/Nb:SrTiO₃ junction" is now accepted for publication in Royal Society Open Science.

on behalf of Dr Chong Li (Associate Editor) and Professor Miles Padgett (Subject Editor)
openscience@royalsociety.org

Appendix A

Reviewer reports:

Response:

We greatly appreciate that the reviewer gave us so many valuable suggestions for improving our manuscript. The manuscript has been revised carefully based on the comments and suggestions.

Reviewer #1: (1) While the mechanism of migration of oxygen vacancies is plausible, its effect on the current through the pn junction ought to be clarified. Either vacancies change the density of minority carriers by increasing the barrier height in which case their effect should be discussed in relation to the schockley equation for the pn junction Or the current increases through the formation of deep O2 levels in the gap of the pn junction increasing the leakage current. Which of these two effects is involved? Why is one material p-doped and the other n-doped?

Response:

CoO is usually a p-type semi-conductor due to the existence of Co vacancies [1], while Nb:SrTiO₃ substrate is a n-type semi-conductor because of the Nb doping. The effect of migration of oxygen vacancies is indeed related to the schockley equation of the pn junction. We added the nonlinear fittings of the transport characteristics in Fig. 1 to analyze the effect of migration of oxygen vacancies on the current through the pn junction, which was uploaded in the revised manuscript.

The revised Fig. 1 is shown as follows:

Figure 1 (a) The schematic of heterojunction CoO/Nb:SrTiO₃. (b) I-V characteristic showing non-volatile bipolar memristive switches. Arrows indicate the voltage-sweep direction. Five cycles of operation are shown. (c) $\ln(I/V)$ vs $(V)^{1/2}$ in the negative bias region at HRS. (d) $\ln(I)$ vs $\ln(V)$ in the negative bias region at LRS. (e)(f) The schematic diagram of the mechanism of RS behaviors in CoO/Nb:SrTiO₃ heterostructure. (e) At forward bias voltage (LRS) and (f) at reverse bias voltage (HRS). (g) Analogy between the CoO/Nb:SrTiO₃ based device and the biological synapse.

And the following sentences were added in page 2 lines 69-73. It reads:

To investigate the origin of the switching behavior, we also carried out nonlinear

fittings to the transport characteristics. Figure 1(c) shows an $\ln(I/V)$ versus $V^{1/2}$ plot, indicating the Poole–Frenkel(P–F) emission mechanism at HRS. However, Figure 1(d) shows an approximate linear relationship between $\ln(I)$ and $\ln(V)$, showing space charge limited (SCL) conduction at LRS. This behavior is consistent with previous reports on the interfacial effects, where the migration of oxygen vacancies and charge trapping/detrapping in the vicinity of the interface drives RS in various heterojunctions [26-28].

(2) In multiple places there is confusion between “current”, “conductance”, “resistance” and “synaptic weight”. For example in Fig 1 when $V > 0$ the current increases with V but the conductance decreases (due to the sub-linear current responses). Another example in Fig 4, the relation between “weight” and “conductance” should be clarified as the conductance is always positive but the weight is negative. Please define what you mean by weight.

Response: Sorry for the confusion. We have revised several places in the paper in order to clarify the conceptual confusion between “current”, “conductance”, “resistance” and “synaptic weight”.

First, we have rewritten the description for Figure 1(b) in order to demonstrate that here we only consider the resistance change driven by the applied voltage and therefore the resistance here is inversely proportional to current shown in the Figure. Here we also added the following sentences to explain why the memristive characteristic is suitable for mimicking biological synaptic weight modulation.

In page 2 lines 60-68, it reads:

Figure 1(b) presents the current-voltage (IV) characteristics of CoO/NSTO in semilogarithmic scale. It is measured by applying a sweeping voltage of $0V \rightarrow +1.5V \rightarrow 0V \rightarrow -2V \rightarrow 0V$. The arrows in Figure 1(b) indicate the sweeping direction. The device resistance can be set to high resistance state (HRS) to low resistance state (LRS) under a positive voltage and reset to HRS under a negative voltage. Besides, the device shows a typical non-volatile bipolar memristive switching performance without a forming process. The memristive characteristic is suitable for mimicking biological synaptic weight modulation for two reasons. First, the resistance variation induced by voltage stimuli is similar to the synaptic weight modification responding to the firing of pre-and post-synaptic neurons. Second, bipolar switching avoids abrupt resistance transition, so that it supports analog-like transition between the resistance states. This is the base of achieving continuously change in synaptic weight.

Second, we focused on emulating the biological excitatory synapse in this paper because phenomena such as long-term potentiation, long-term depression, usually occurs at excitatory synapses [2]. An excitatory synapse increases the probability of an action potential occurring in post-synaptic neurons and therefore the weight, or the connection strength of an excitatory synapse is always positive. However, the change of the synaptic weight can be either positive or negative. The x-axis in Figure 4 was showing the weight change of the electrical synapse, which was represented by the

change of the device conductance. And in order to explain what “synapse” and “synaptic weight” refers to in this paper as well as clarify the relation between “synaptic weight” and “device resistance/conductance”, we have added the following sentences:

In page 2 lines 84-93 it reads:

Figure 1(g) illustrates an analogy between the electronic synapse based on CoO/Nb:SrTiO₃ memristor and the biological excitatory synapse. Note that in this paper we specifically focus on emulating excitatory synapses because phenomena such as long-term potentiation, long-term depression, usually occurs at these synapses [29]. As shown in Figure 1(e), a biological synapse is composed of a pre-synaptic neuron, a post-synaptic neuron, and a synapse. In electronic synapse based on CoO/Nb: SrTiO₃ memristor, the negative and positive electrodes work as the pre- and post-synaptic neurons, respectively. The synaptic weight, which reflects the connection strength between two neighbouring neurons, is represented by the device resistance or conductance. The resistance or conductance of the electronic device is changed because of the oxygen vacancies migration when receiving the input action potentials. This is similar to the biological synaptic weight, which is changed because the pre-neuron releases ions when receiving the input spikes. And the device resistance is inversely proportional to the synaptic weight while the device conductance is proportional to the synaptic weight.

In page 3 lines 146-148, it reads:

Here we use the conductance change of the device to represent the synaptic weight modulation. Although the conductance is always positive, the change of it can either be positive or negative.

Third, we corrected “resistance” to “conductance” in the following lines:

Page 3 lines 153-154: **.In order to exclude *conductance* modification by a single spike, the pulse and tail voltage amplitudes were selected to be both lower than the threshold voltage V_{th} for SET and RESET processes.**

Page 4 line 165: **Figure 4 shows the resulting timing-dependent *conductance* modulation.**

Page 4 lines 166-168: **For high Δt , the magnitude of injected voltage does not exceed V_{th} at any time, thus *conductance* remains unchanged. For low Δt , however, the magnitude of injected voltage in excess of V_{th} is applied, *conductance* is thus increased or decreased depending on the sign of Δt .**

Minor comments:

- A horizontal timescale in the inset of Fig 3. Would be useful.

Response:

Thanks for the suggestion.

The inset of Fig. 3 was a diagram of the pulse scheme. We have revised the figure and added a horizontal timescale. Now the inset of Fig. 3 is displaying the actual time for the horizontal axis.

The revised Fig. 3 is shown as follows:

Figure 3 Resistance switching cycles driven by consecutive positive or negative pulses, representing synaptic weight modulation due to the potentiating or depressing pulses, respectively. Upper inset: the pulse schemes. The pulse amplitudes vary with identical 2 ms widths and 1 s pulse intervals.

- The description of voltage pulses used to demonstrate STDP action in Fig 4 is not described sufficiently clearly in the text or the caption to allow the experiment for be replicated. As these are not square pulses, explain what the $t=50\text{ms}$ and $t=380\text{ms}$ mean.

Response:

Thanks for the suggestion.

We have added the vertical and horizontal scales in insets of Figure 4 in order to show the details of the voltage pulses.

The revised Fig. 4 is shown as follows:

Figure 4 Plot of the device conductance (synaptic weight) change with variation in Δt showing the STDP response of the electronic synapse. Inset: Schematic of the pulse-pair applied to our device for STDP demonstration. Although amplitude of each pulse is under the threshold, the sum of them can exceed the threshold of SET and RESET process and lead to change in device conductance (synaptic weight). When the pre-spike precedes the post-spike ($\Delta t > 0$), the device is set (potentiated); when the post-spike precedes the pre-spike ($\Delta t < 0$), the device is reset (depressed). The timing (Δt) between the two spikes determines the voltage flux ($V \times \Delta t$) across the device.

And we have added the following sentences in order to explain how we set the particular parameters of these voltage pulses.

In page 3 lines 150-153, it reads:

The detailed analysis of the different shapes of “spikes” and the resulting STDP function was discussed in the paper of T. Serrano. Gottaredong et al. [39]. Although the shape of the spike is not decisive, it is assumed that each spike in the pair consists of a narrow short positive pulse of large amplitude and a long relaxing slowly decreasing negative tail.

In page 3 lines 154-157, it reads:

And we arbitrary set the duration of the short positive pulse and the duration of the tail voltage to be 50 ms, and 380 ms, respectively. In this way, the spike has the shape very close to those in biological systems. Meanwhile, this setting leads to stimulation that is great enough to induce the conductance variation.

In page 3 line 161 and page 4 lines 162-164, it reads:

When the two spikes with opposite polarities arrived at the device, voltages

added up. The amplitude of the total voltage at the device could exceed the threshold voltage V_{th} , leading a conductance variation. As Δt increases, both of the effective voltage and the effective time decrease, eventually leading to the decrease in synaptic weight change, and vice versa.

Reviewer #2: (1) It is said that the migration of oxygen vacancies and charge trapping and detrapping at the heterojunction interface, which modify the depletion layer width and the interface barrier height, is responsible for the resisting switching behavior in the CoO/Nb: SrTiO₃ junctions. The reviewer was not sure from which experimental data the author concluded this proposed mechanism.

(2) The schematic in Fig. 1e indicates that oxygen vacancies could move from CoO layer' surface on the positive electrode side to the Nb-STO layer' s surface on the negative electrode. Is there any experimental evidences on the oxygen ion movements in the CoO/Nb-STO junctions? The reviewer was not sure how the authors verify the schematic. More explanations will be necessary.

Response to comment (1) and comment (2):

We accepted the reviewer's comments and suggestions. The manuscript has been revised carefully based on these comments and suggestions. We added the nonlinear fittings of the transport characteristics in Fig. 1 to investigate the origin of the resistive switching behaviors, which was uploaded in the revised manuscript. The following sentence was added in page 2 lines 69-73. It reads:

To investigate the origin of the switching behavior, we also carried out nonlinear fittings to the transport characteristics. Figure 1(c) shows an $\ln(I/V)$ versus $V^{1/2}$ plot, indicating the Poole–Frenkel (P–F) emission mechanism at HRS. However, Figure 1(d) shows an approximate linear relationship between $\ln(I)$ and $\ln(V)$, showing space charge limited (SCL) conduction at LRS. This behavior is consistent with previous reports on the interfacial effects, where the migration of oxygen vacancies and charge trapping/detrapping in the vicinity of the interface drives RS in various heterojunctions [26-28].

The revised Fig. 1 is shown as follows:

Figure 1 (a) The schematic of heterojunction CoO/Nb:SrTiO₃. (b) I-V characteristic showing non-volatile bipolar memristive switches. Arrows indicate the voltage-sweep direction. Five cycles of operation are shown. (c) $\text{Ln}(I/V)$ vs $(V)^{1/2}$ in the negative bias region at HRS. (d) $\text{Ln}(I)$ vs $\text{Ln}(V)$ in the negative bias region at LRS. (e)(f) The schematic diagram of the mechanism of RS behaviors in CoO/Nb:SrTiO₃ heterostructure. (e) At forward bias voltage (LRS) and (f) at reverse bias voltage (HRS). (g) Analogy between the CoO/Nb:SrTiO₃ based device and the biological synapse.

(3) The authors need to add more explanations on details of their measurements. Especially measurements of the junction resistance changes induced by applying pulse voltages. How did the authors measure/probe the resistance changes after applying pulse voltages?

Response:

Thanks for the suggestion. We have added the following sentence in order to explain the details of our measurements, especially the measurements of the junction resistance changes induced by applying pulse voltages.

In page 2 lines 96-97, it reads:

The device resistance was monitored by applying a read voltage of 100mV for 100 ms after each pulse stimulus.

In page 3 lines 108-111, it reads:

The time intervals between pulses were 1 s (900 ms waiting time plus 100 ms read time). During this interval, the resistance state was monitored by a read voltage of 100 mV.

In page 3 lines 129-130, it reads:

All pulse widths were fixed at 2ms. Each pulse was followed by measurement of the device resistance.

(4) For measurements in Figure 2 and 3, how long intervals between pulses were used? If the interval would be changed, how would the junction resistance behavior be influenced? The authors are asked to add more descriptions. If it is the case that the intervals affect the junction resistance behavior, is it still safely say that the Co/Nb-STO junction can emulate synapse properties?

Response:

The intervals between pulses would not affect the junction resistance behavior because of the nonvolatile memory of the junction, which refers to a stable resistance state after being stimulated. Therefore it is safely say that the Co/Nb-STO junction can emulate synapse properties. In order to demonstrate the interval between pulses in Figure 2 and 3, we have added the following sentences:

In page 3 lines 108-111, it reads:

The time intervals between pulses were 1 s (900 ms waiting time plus 100 ms read time). During this interval, the resistance state was monitored by a read voltage of 100 mV. The time interval between pulses would not affect the device resistance because of the nonvolatile memory of the device, which refers to a stable resistance state after being stimulated.

In page 3 lines 129-131, it reads:

All pulse widths were fixed at 2ms. Each pulse was followed by measurement of the device resistance. Note that the time interval between pulses would not affect the device resistance because of the non-volatile memory of the device.

Reference:

- [1] Higgs S, Eskenazi T. Resistance switching in the metal deficient-type oxides: NiO and CoO. *Applied Physics Letters* 2007;91:5655-181.
- [2] Gerrow K, Triller A. Synaptic stability and plasticity in a floating world. *Current Opinion in Neurobiology* 2010;20:631-9.

Appendix B

Dear Dr. Chong Li, Editor

I should like to express my appreciation to you for suggesting how to improve our manuscript. The manuscript has been revised carefully based on your comment and suggestion.

Associate Editor Comments to Author (Dr Chong Li):

Thank you for revising the paper after considering our fellow reviewers' comments. However one of our reviewers is still concerned about the real mechanism of the resistance switching phenomenon. Although the authors tried to use curve-fitting to explain it with relevant reference support in the reply, it is still not convincing. It would be better if this can be experimentally demonstrated as shown in Ref. 26.

Response:

The release process of low resistance states was fitted well with a bi-exponential relaxation equation as shown in the inset of Figure 1(d), which can be attributed to the oxygen vacancy movements. In the revised manuscript, we added the following sentences to discuss the double-exponential decay:

In page 2 lines 74-77, it reads: **“Moreover, the time dependence of the current at the LRS was fitted with a bi-exponential relaxation equation of $I = I_0 + Ae^{-t/\tau_1} + Be^{-t/\tau_2}$, where I_0 is the steady state current, t is the time, A and B are constant, and τ_1 and τ_2 are two relaxation time constants. The well fitted curve with $\tau_1 = 27.7\text{s}$ and $\tau_2 = 337.6\text{s}$ further implies oxygen vacancy migration.”**

We added the time dependence of the current at the LRS and the fitting curve in the inset of Figure 1(d). The revised Figure 1 is shown as follows:

Figure 1 (a) The schematic of heterojunction CoO/Nb:SrTiO₃. (b) I-V characteristic showing non-volatile bipolar memristive switches. Arrows indicate the voltage-sweep direction. Five cycles of operation are shown. (c) $\text{Ln}(I/V)$ vs $(V)^{1/2}$ in the negative bias region at HRS. (d) $\text{Ln}(I)$ vs $\text{Ln}(V)$ in the negative bias region at LRS. Inset: the time dependence of the current at the LRS and the fitting curve. (e)(f) The schematic diagram of the mechanism of RS behaviors in CoO/Nb:SrTiO₃ heterostructure. (e) At forward bias voltage (LRS) and (f) at reverse bias voltage (HRS). (g) Analogy between the CoO/Nb:SrTiO₃ based device and the biological synapse.

Reviewer: 2

Comments to the Author(s)

The reviewer thanks authors for their efforts on revising the manuscript and responding comments/suggestions. However, the reviewer is still not sure if migrations of oxygen vacancies are the main origin for the resistance switching behavior of the CoO/Nb-STO junctions. Although the authors are claiming that this conclusion is supported by the non-linear analysis on the IV characteristics of the junctions, the authors' analysis is indirect and the oxygen vacancy migration scenario is just one of possibilities. This is a critical point because some discussion in this manuscript are based on this claim. The reviewer thinks that the author should provide direct evidences on this point.

Response:

Thanks for your review. In transition metal oxides, electrochemical migration of oxygen ions has been widely regarded as the driving mechanism of bipolar-type resistive switching [1]. However, to the best of our knowledge, the direct observation of oxygen vacancy movements during resistive switching process has not been reported. Oxygen vacancy movements are usually indirectly demonstrated by IV curve fitting in Ref. 26 or magnetism variation [2]. Since defects at the heterojunction interface play a significant impact on the conduction mechanism. The change of the IV curve implies variations in the height and width of the barrier, which can be attributed to the change of the defect distribution. At the same time, time relaxation of resistance states can also be related with the oxygen vacancy movement [26].

Based on the editor's suggestion, we added the well fitted curve of the double-exponential decay in the inset of Figure1(d) and the following sentences in the revised manuscript:

In page 2 lines 74-77, it reads: **“Moreover, the time dependence of the current at the LRS was fitted with a bi-exponential relaxation equation of $I = I_0 + Ae^{-t/\tau_1} + Be^{-t/\tau_2}$, where I_0 is the steady state current, t is the time, A and B are constant, and τ_1 and τ_2 are two relaxation time constants. The well fitted curve with $\tau_1 = 27.7s$ and $\tau_2 = 337.6s$ further implies oxygen vacancy migration.”**

The revised Figure 1 is shown as follows:

Figure 1 (a) The schematic of heterojunction CoO/Nb:SrTiO₃. (b) I-V characteristic showing non-volatile bipolar memristive switches. Arrows indicate the voltage-sweep direction. Five cycles of operation are shown. (c) $\text{Ln}(I/V)$ vs $(V)^{1/2}$ in the negative bias region at HRS. (d) $\text{Ln}(I)$ vs $\text{Ln}(V)$ in the negative bias region at LRS. Inset: the time dependence of the current at the LRS and the fitting curve. (e)(f) The schematic diagram of the mechanism of RS behaviors in CoO/Nb:SrTiO₃ heterostructure. (e) At forward bias voltage (LRS) and (f) at reverse bias voltage (HRS). (g) Analogy between the CoO/Nb:SrTiO₃ based device and the biological synapse.

Reference:

- [1] Sawa A. Resistive switching in transition metal oxides. *Materials Today* 2008;11:28-36.
- [2] Li Q, Yan SS, Xu J, Li SD, Zhang J. Electrical control of exchange bias via oxygen migration across CoO-ZnO nanocomposite barrier. *Applied Physics Letters* 2016;109:252406.